# Impact of decreasing the proportion of higher energy foods and reducing portion sizes on food purchased in worksite cafeterias: A stepped-wedge randomised controlled trial

**James P. Reynolds**[1,2]*, **Minna Ventsel**[1], **Daina Kosīte**[1], **Brier Rigby Dames**[1], **Laura Brocklebank**[3], **Sarah Masterton**[4], **Emily Pechey**[1], **Mark Pilling**[1], **Rachel Pechey**[1,5], **Gareth J. Hollands**[1], **Theresa M. Marteau**[1]*

1 Behaviour and Health Research Unit, University of Cambridge, Cambridge, United Kingdom, 2 School of Psychology, Aston University, Birmingham, United Kingdom, 3 School of Psychological Science, University of Bristol, Bristol, United Kingdom, 4 Department of Psychological Sciences, University of Liverpool, Liverpool, United Kingdom, 5 Nuffield Department of Primary Care and Health Science, University of Oxford, Oxford, United Kingdom

* j.reynolds4@aston.ac.uk (JPR); tm388@medschl.cam.ac.uk (TMM)

**Data Availability Statement:** This study includes sales data from the collaborating supermarket,

## Abstract

### Background

Overconsumption of energy from food is a major contributor to the high rates of overweight and obesity in many populations. There is growing evidence that interventions that target the food environment may be effective at reducing energy intake. The current study aimed to estimate the effect of decreasing the proportion of higher energy (kcal) foods, with and without reducing portion size, on energy purchased in worksite cafeterias.

### Methods and findings

This stepped-wedge randomised controlled trial (RCT) evaluated 2 interventions: (i) availability: replacing higher energy products with lower energy products; and (ii) size: reducing the portion size of higher energy products. A total of 19 cafeterias were randomised to the order in which they introduced the 2 interventions. Availability was implemented first and maintained. Size was added to the availability intervention. Intervention categories included main meals, sides, cold drinks, snacks, and desserts. The study setting was worksite cafeterias located in distribution centres for a major United Kingdom supermarket and lasted for 25 weeks (May to November 2019). These cafeterias were used by 20,327 employees, mainly (96%) in manual occupations. The primary outcome was total energy (kcal) purchased from intervention categories per day. The secondary outcomes were energy (kcal) purchased from nonintervention categories per day, total energy purchased per day, and revenue. Regression models showed an overall reduction in energy purchased from intervention categories of −4.8% (95% CI −7.0% to −2.7%), $p < 0.001$ during the availability intervention period and a reduction of −11.5% (95% CI −13.7% to −9.3%), $p < 0.001$ during the availability plus size intervention period, relative to the baseline. There was a reduction in

which were received weekly from May 2019 to November 2019. Due to contractual restrictions to the use of the sales data, these data cannot be made openly available. Any requests to access the data can be directed to our administrators (bcbd.administrator@medschl.cam.ac.uk). The collaborating supermarket wished to remain anonymous and any requests for further use of these data must be sent via our administrators who will send the request to the collaborating supermarket.

**Funding:** This research was funded in whole, or in part, by the Wellcome Trust [ref: 206853/Z/17/Z (TMM, GJH)]. For the purpose of Open Access, the author has applied a CC BY public copyright licence to any Author Accepted Manuscript version arising from this submission. CRUK (Cancer Research UK) [ref: C4770/A29425- Health Evaluation Targeted Research (TMM)] and the Wellcome Trust [ref: 106679/Z/14/Z (RP)] also contributed funds towards this research. The funders had no role in study design, data collection and analysis, decision to publish, or preparation of the manuscript.

**Competing interests:** The authors have declared that no competing interests exist.

**Abbreviations:** BMI, body mass index; CONSORT, Consolidated Standards of Reporting Trials; GLMM, generalised linear mixed model; Q–Q, quantile–quantile; RCT, randomised controlled trial; TIPPME, Typology of Interventions in Proximal Physical Micro-Environments.

energy purchased of −6.6% (95% CI −7.9% to −5.4%), $p < 0.001$ during the availability plus size period, relative to availability alone. Study limitations include using energy purchased as the primary outcome (and not energy consumed) and the availability only of transaction-level sales data per site (and not individual-level data).

## Conclusions

Decreasing the proportion of higher energy foods in cafeterias reduced the energy purchased. Decreasing portion sizes reduced this further. These interventions, particularly in combination, may be effective as part of broader strategies to reduce overconsumption of energy from food in out-of-home settings.

## Trial registration

ISRCTN registry ISRCTN87225572.

---

## Author summary

### Why was this study done?

- Overconsumption of energy from food and drink contributes to the high and rising rates of obesity in many countries.
- Creating healthier food environments is the key to enabling healthier diets.
- There is insufficient real-world evidence on the best ways to improve food environments.

### What did the researchers do and find?

- Two interventions were evaluated: (i) availability: replacing some higher energy foods with lower energy foods; and (ii) size: reducing the portion size of some higher energy foods.
- We carried out a randomised stepped-wedge trial in 19 cafeterias spanning 25 weeks.
- The availability reduced energy purchased by 4.8%, and the availability plus size intervention reduced energy purchased by 11.5%.

### What do these findings mean?

- Replacing some higher energy foods in cafeterias with lower energy options and reducing the portion size of some higher energy foods both appear to be effective strategies for reducing energy purchased in worksite cafeterias.
- These interventions can contribute to broader strategies to reduce energy intake out of the home, as part of national and international efforts to tackle overweight and obesity.

## Introduction

Unhealthy patterns of food consumption, including excess energy intake, are a major contributor to high and rising rates of obesity, leading to an increased prevalence of noncommunicable diseases and premature death worldwide [1–3]. These increasing rates of obesity are attributable to the environments in which we live, which influence the purchasing and consumption of food and drinks [4,5]. Local areas of deprivation also magnify this effect; people living in deprived areas or with lower socioeconomic status tend to have reduced access to healthy foods and higher rates of obesity [6,7]. Targeting unhealthy food environments for intervention is therefore one of the most promising strategies for reducing obesity and its associated health and economic burden [8].

One important environment where interventions could be implemented is cafeterias, such as those in schools, universities, and workplaces. The workplace is the most common place to eat outside of the home, with 14% to 17% of adults' energy intake occurring while at work [9]. The recent UK obesity strategy also highlights the workplace as a key place in which to encourage healthier eating [10]. There are many interventions that alter such food environments, and a recent Typology of Interventions in Proximal Physical Micro-Environments (TIPPME) [11] describes 6 of these interventions, each of which can be applied to products (e.g., the food itself), related objects (e.g., the table on which the food is served or consumed), or the wider environment (e.g., the cafeteria's structure or layout). The interventions described in this typology overlap with the more general concept of "nudging," but the typology aims to provide more precisely operationalised classifications of different intervention types. Two Cochrane reviews have highlighted the potential for 2 types of these interventions when applied to products: decreasing the relative availability of higher energy products [12] and reducing portion sizes of some higher energy products [13]. Despite evidence for effectiveness at changing eating behaviours, both reviews highlighted the same problems: a lack of real-world studies evaluating the interventions, the limited size of the few studies that were conducted in real-world settings, and an overreliance on multicomponent interventions that preclude causal inferences regarding the effectiveness of individual components. One study identified in the review on availability interventions was a study in 6 worksite cafeterias, which evaluated the effect of replacing some higher energy products with lower energy alternatives, leaving the same overall number of options [14]. This intervention reduced the amount of energy (kcals) purchased in the cafeteria by approximately 7%. One study identified in the review on portion size took place in 6 worksite cafeterias and evaluated the effect of reducing portion sizes of selected products by at least 10% [15]. The results showed reductions in energy purchased of approximately 9%, although this reduction was not statistically significant. Taken together, these interventions show promise and warrant further, adequately powered research.

The current study—the largest of its kind—to our knowledge—aimed to build on these earlier studies by testing the impact of both availability and size interventions in a larger number of worksite cafeterias, over a longer period of time, to determine the robustness of these original studies. We also aimed to investigate the impact of implementing an availability intervention singly, as well as in combination with a size intervention, to estimate the individual and combined effectiveness.

## Methods

The study was prospectively registered on ISRCTN (ISRCTN87225572), and a detailed analysis plan was uploaded to the Open Science Framework (https://osf.io/vyze4/) during data collection, but before data cleaning or analysis had commenced. Further details of the study are provided in the published protocol [16]. The Consolidated Standards of Reporting Trials

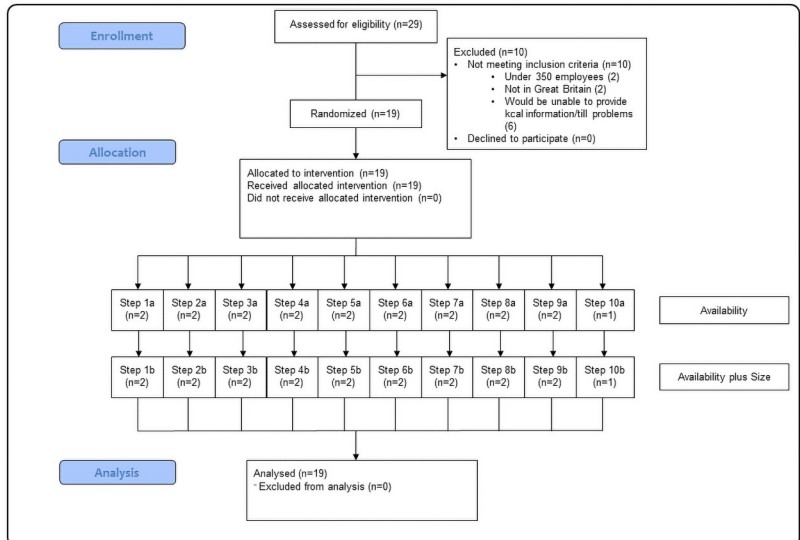

**Fig 1. Study site flowchart.** Note: *n* refers to cafeterias. Each step refers to weekly periods in which cafeterias start the interventions. See Fig 2 for the timings of these steps.

(CONSORT) extension checklist for stepped-wedge trials is attached as a Supporting information (see S1 CONSORT Checklist). The Cambridge Psychology Research Ethics Committee based at the University of Cambridge approved the trial on May 14, 2019 (No. PRE.2019.006). The research team obtained written informed consent from a representative of the supermarket group on behalf of the participating sites.

## Sites

A total of 19 worksite cafeterias were recruited through a major UK supermarket chain (see Fig 1) and were based within their distribution centres. The distribution centres were typically in remote locations with no local food outlets to purchase food or drink. Other than the cafeteria—the location in which we intervened—the only options were vending machines (containing snacks and drinks) and bringing in food from home. There were 3 eligibility criteria: They needed (i) to be located in Great Britain; (ii) to have at least 350 employees based at the site; and (iii) to have electronic point-of-sale tills to record sales data. A total of 29 cafeterias were screened for eligibility, and 19 participated in the study (Fig 1). The participating sites housed 1 cafeteria each and employed between 561 and 2,357 staff (20,327 in total), most of whom were employed in manual jobs (see Table 1). The participating cafeterias were managed by 3 catering companies: one company managed 10 cafeterias, one managed 5, and one managed 4.

The sample size calculation was based on data from 2 similar studies [14,15]. We conservatively used the largest estimate of SD (0.111) when analysis was on the log scale. A one-sided *t* test at 80% power with 5% significance level would require *n* = 19 cafeterias to detect an effect size of 6.5% or greater reduction in energy purchased. We selected a one-sided test, reflecting existing evidence [12–15].

## Intervention periods

### Baseline

The baseline period, during which no intervention was implemented, was mostly a period of business as usual for the cafeterias. Prior to taking part in the study, cafeterias would respond

**Table 1. Demographic characteristics of employees in participating sites.**

| Cafeterias | Number of employees (mean) | Proportion male (%) | Mean age (SD) | Proportion nonmanagerial workers (%) |
|---|---|---|---|---|
| Total | 20,327 (1,070) | 85 | 39 (12) | 96 |
| 1 | 813 | 89 | 44 (14) | 95 |
| 2 | 1,098 | 80 | 35 (10) | 97 |
| 3 | 561 | 67 | 41 (12) | 94 |
| 4 | 749 | 83 | 37 (12) | 96 |
| 5 | 605 | 85 | 43 (12) | 92 |
| 6 | 1,403 | 85 | 36 (12) | 97 |
| 7 | 1,105 | 86 | 41 (13) | 97 |
| 8 | 758 | 91 | 37 (12) | 96 |
| 9 | 1,001 | 83 | 40 (13) | 97 |
| 10 | 2,357 | 87 | 34 (10) | 98 |
| 11 | 589 | 82 | 41 (13) | 96 |
| 12 | 837 | 81 | 43 (13) | 96 |
| 13 | 1,348 | 74 | 35 (10) | 97 |
| 14 | 1,497 | 85 | 38 (12) | 97 |
| 15 | 1,456 | 88 | 39 (13) | 97 |
| 16 | 774 | 88 | 43 (12) | 96 |
| 17 | 926 | 93 | 45 (13) | 96 |
| 18 | 1,644 | 86 | 42 (12) | 97 |
| 19 | 806 | 92 | 43 (14) | 96 |

to surpluses or shortfalls in certain ingredients by changing which meals would be offered or by altering recipes. This flexibility was restricted during the study, and cafeterias were instructed to use fixed menus and recipes from the beginning of the baseline period to ensure that energy content could be accurately estimated for all meals. There were 7 recorded instances of a meal deviating from the planned meal on that day. When this occurred, the new meal's energy content was used and not that of the planned meal.

## Availability

The availability intervention was implemented in isolation (i.e., without size) during the availability period. In the current study, this intervention comprised decreasing the number of higher energy food and drink products and increasing the number of lower energy food and drink products to maintain the same total number of products, i.e., changing the relative availability [17]. In the TIPPME [11], this is classified as an availability × product intervention.

Cutoff points in energy content by product type were prespecified to identify products that could be removed and those that could be added (see Table A in S1 Text). The planned degree of implementation—such as reducing higher energy sweet snacks from 90% to 80%—was negotiated with the catering providers and was primarily determined by their ability to procure lower energy options. It was possible to apply the availability intervention to 7 food and drink categories, which made up 54% of all energy purchased:

1. main meals (e.g., lasagna);

2. side dishes (e.g., rice);

3. cold drinks (e.g., can of cola);

4. sweet snacks (e.g., chocolate bar);

**Table 2. Intervention characteristics.**

| Cafeteria | Proportion (%) of intervention category products that were higher energy per day during baseline | Proportion (%) of intervention category products that were higher energy per day during availability | Proportion (%) of intervention category products that were higher energy per day during availability plus size | Availability implementation (%) across all intervention categories | Proportion (%) of intervention category products on sale per day reduced in size | Mean reduction in size by volume (%) | Size implementation (%) across all intervention categories |
|---|---|---|---|---|---|---|---|
| Overall | 58 (49/85) | 50 (42/84) | 47 (38/82) | 76 | 7 (6/82) | 14 | 83 |
| 1 | 64 (46/71) | 59 (47/80) | 58 (49/85) | 76 | 8 (6/85) | 12 | 74 |
| 2 | 52 (48/92) | 41 (32/80) | 39 (31/80) | 96 | 4 (3/80) | 17 | 87 |
| 3 | 61 (53/87) | 53 (46/86) | 48 (43/89) | 93 | 2 (2/89) | 10 | 75 |
| 4 | 63 (41/65) | 60 (36/59) | 55 (29/53) | 58 | 7 (3/53) | 17 | 75 |
| 5 | 61 (44/73) | 49 (35/72) | 44 (36/82) | 75 | 2 (2/82) | 10 | 100 |
| 6 | 60 (94/157) | 48 (59/124) | 44 (47/106) | 75 | 4 (4/106) | 17 | 81 |
| 7 | 61 (44/73) | 51 (40/78) | 48 (41/85) | 68 | 2 (2/85) | 10 | 70 |
| 8 | 67 (59/88) | 65 (47/73) | 56 (50/89) | 78 | 3 (3/89) | 10 | 100 |
| 9 | 61 (32/53) | 58 (32/55) | 52 (27/52) | 79 | 17 (8/52) | 12 | 100 |
| 10 | 43 (35/80) | 36 (45/125) | 29 (32/112) | 89 | 5 (6/112) | 16 | 100 |
| 11 | 64 (44/69) | 52 (34/66) | 50 (40/80) | 74 | 10 (8/80) | 12 | 78 |
| 12 | 59 (44/74) | 47 (36/77) | 51 (38/75) | 55 | 5 (3/75) | 16 | 75 |
| 13 | 62 (45/73) | 54 (42/77) | 51 (42/82) | 89 | 2 (2/82) | 10 | 100 |
| 14 | 54 (68/126) | 47 (45/95) | 47 (39/83) | 89 | 4 (3/83) | 16 | 100 |
| 15 | 52 (71/137) | 54 (80/148) | 46 (51/110) | 81 | 3 (4/110) | 16 | 69 |
| 16 | 47 (8/16) | 36 (11/30) | 41 (16/39) | 74 | 10 (4/39) | 16 | 58 |
| 17 | 57 (37/64) | 46 (24/52) | 44 (21/48) | 50 | 8 (4/48) | 17 | 75 |
| 18 | 56 (86/153) | 50 (72/144) | 46 (70/152) | 82 | 3 (4/152) | 16 | 59 |
| 19 | 59 (40/68) | 53 (44/82) | 52 (33/64) | 67 | 7 (4/64) | 13 | 100 |

5. savoury snacks (e.g., crisps);

6. desserts (e.g., pot of chocolate mousse); and

7. bakery items (e.g., slice of Victoria sponge cake).

Further categories that were not targeted included breakfasts, soups, salads, and sandwiches, as it was not possible to intervene in these categories. Sandwiches were initially planned to be targeted for the availability intervention; however, it was subsequently discovered that there were contracts that prevented removing sandwiches from sale. The planned implementation was based on our pilot study [14]. The intervention characteristics are shown in Table 2 and S1 and S2 Data.

## Availability plus size

For the availability plus size period, the availability intervention was maintained, and the size intervention was implemented. The size intervention comprised reducing the portion size, by volume, of products in targeted food categories. Within the TIPPME intervention typology [11], this is classified as a size × product intervention.

It was possible to apply the size intervention to 4 of the 7 categories described above:

1. main meals;

2. sides;

3. desserts; and

4. bakery items.

Changes were made only to products in those 4 categories that were classified as higher energy using the cutoffs in Table A in S1 Text. The planned reductions in portion size varied by cafeteria and product, but were requested to be at least a 10% reduction in volume for each targeted product. We also requested that any reduction in portion size was accompanied by a proportionate change in price to avoid confounding a change in size with a change in value for money. This request to reduce the price was not granted as the products were already subsidised.

## Design

A stepped-wedge design was used across a period of 25 weeks (27.05.19 to 18.11.19). Such designs are typically preferred to a parallel groups randomised controlled trial (RCT) when study resources only allow a staggered implementation of the intervention(s). Each of the 19 cafeterias was randomly allocated to the week in which the intervention was implemented (see Fig 2). The baseline period lasted between 4 and 13 weeks, depending on the randomisation order of each cafeteria. Weeks 1 to 4 comprised the minimum baseline period. From Week 5 until Week 13, 2 cafeterias a week implemented the first intervention—availability. There was no gap between steps; interventions were implemented on the first day of the intervention periods. In Week 14, the last cafeteria (number 19) implemented the availability intervention. The availability intervention period lasted 8 weeks for all cafeterias. From Week 13 until Week 21, 2 cafeterias a week implemented the second intervention—size—while continuing the availability intervention. In Week 22, the last cafeteria implemented the size intervention. The interventions continued for all cafeterias until the end of Week 25, as planned. The availability plus size intervention period, therefore, lasted 4 to 13 weeks, depending on the randomisation order of each cafeteria.

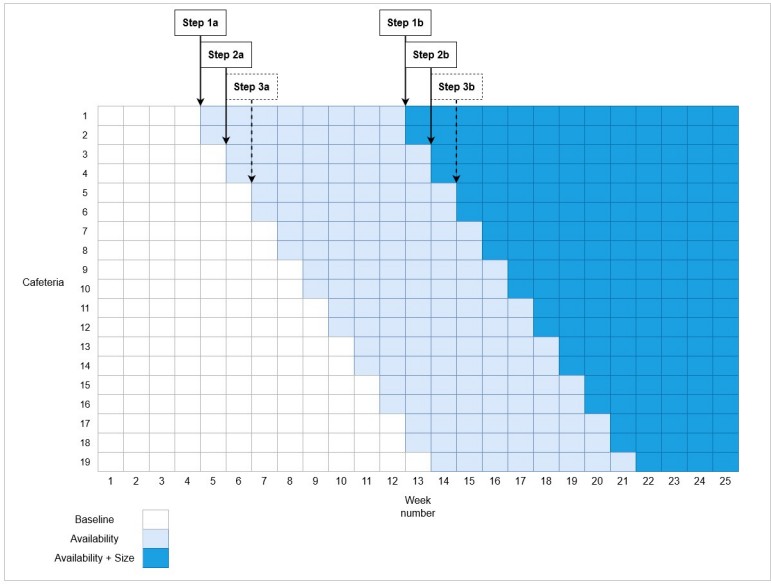

**Fig 2. Stepped-wedge study design.**

While, ideally, both availability and size would have been investigated individually as well as together, due to power constraints, we needed to select one. We opted for availability for 2 reasons: (i) the pilot study favoured the availability intervention, and we wanted to test the more promising intervention in isolation; and (ii) it made more sense to reduce portion sizes after changing products, as opposed to reducing portion sizes first, and then removing products.

## Randomisation and masking

Participating sites were randomly allocated to the time in which the interventions were implemented. The randomisation was performed by a statistician who was blinded to the identity of the sites. The statistician allocated a list of anonymised site names using the rank of random numbers from Excel. The customers of each cafeteria were not individually informed of the study, but posters were displayed in the cafeterias, and internal communications circulated. These described a new "health initiative" designed to improve the food offering in the cafeterias; however, the specific changes were not described. The dates in which the changes would be implemented were not communicated to customers. Blinding the catering staff to the interventions was not possible as they were implementing the interventions. Sites were informed about their allocation (the week in which they were to implement the interventions) after recruitment and before data collection, which allowed time for them to prepare for the interventions.

## Procedure

Initial cafeteria visits from the research team took place to assess eligibility, supplemented by checks of the full product lists obtained from each catering provider.

**Implementation.** The availability and size interventions were implemented by catering staff at each cafeteria following training and assistance from representatives from the catering companies and research team. For the availability intervention, the cafeteria staff were provided with information on the target proportion of lower energy items per intervention category, along with the definition of lower energy (e.g., cold drinks: 50% of cold drinks available for sale must be below 50 kcals).

For the size intervention, cafeteria staff were advised that the required portion size reduction for targeted products was at least 10%. Items in trays (e.g., lasagna) were reduced by producing more portions from the same tray (10 portions versus 8). Countable items (e.g., meatballs) were reduced accordingly (from 5 to 4). The portion sizes of items that were served using ladles—such as for stew or scoops—such as for chips—were reduced by either instructing staff to use fewer ladles or scoops when serving customers (1.5 scoops versus 2 scoops) or by providing smaller serving equipment (4-oz scoop versus 5-oz scoop). Some cafeterias also weighed foods in individual serving containers to ensure accuracy.

**Fidelity.** Fidelity to protocol was assessed during prearranged visits conducted by the research team on at least 5 occasions per cafeteria throughout the study: 1 during baseline, 2 during availability, and 2 during availability plus size. Every food and drink product sold in the cafeterias was recorded and photographed to assess the implementation of the availability intervention. Fidelity to the size intervention was measured by observing the food and the serving of the food, which was then compared against the agreed portion sizes. Cafeteria staff were also asked what the correct portion sizes should be to ensure that they were aware of the changes.

Any violations to protocol were reported to a manager responsible for the cafeteria with a request to rectify the violation and provide photographic evidence of the rectification within

24 hours. Adherence to or violation of the planned implementation was recorded to be used in secondary analyses.

## Measures

All outcomes were derived from sales data that were recorded during each day of the cafeterias' operation during the study using electronic point-of-sale tills.

**Primary outcome.** This comprised energy (kcal) purchased from intervention food and drink categories per day. This measure was calculated from the total number of sales for all items within an intervention food category and the energy content for each of these items.

Energy content (kcals) was available for most products (98%) on sale at the cafeterias. This information was provided by the 3 catering providers and supplemented by checking labels on certain products. For the remaining items, energy content was estimated using the energy content of target products available from another catering provider, if the product description was the same.

**Secondary outcomes.**

1. Total energy (kcal) purchased per day of analysis from nonintervention categories: These included those not targeted during either intervention: sandwiches, breakfasts, soups, salads, and other foods.

2. Total energy (kcal) purchased from all food and drink products: This included all products within intervention and nonintervention categories.

3. Total revenue from each cafeteria: This was calculated from the total number of items sold in the cafeterias and the price of each of these items.

There were plans to examine sales data from the vending machines; however, it was later found to not be possible to collect these data from the sites.

**Covariates.**

1. Total number of transactions: the number of unique payments to purchase all products in the cafeteria, as a proxy measure for the number of customers per day (for all outcomes except revenue);

2. Time (baseline): day number of the trial within baseline, starting from 1 at each cafeteria;

3. Time (availability): day number within availability, starting from 1 at each cafeteria;

4. Time (availability plus size): day number within availability plus size, starting from 1 at each cafeteria;

5. Day of the week;

6. Catering provider; and

7. Daily mean temperature. Data were accessed from the CEDA Archive [18], and the nearest station to each cafeteria was selected.

## Analyses

Generalised additive linear mixed models [19] were used to estimate the overall potential impact of the availability intervention and the combined availability plus size intervention compared to baseline. Cafeterias were fitted as random effects, with the effect of the day of the week allowed to vary by cafeteria as a random nested term. In additive models, cafeterias had

markedly different variability (heteroscedasticity), which required modelling. Day number during each intervention (normalised within each cafeteria to ensure equal weighting between cafeterias) was used to allow for the fitting of 3 separate overall time trends. The identity link function (i.e., normal distribution) was used for modelling the mean, and the sigma link function (i.e., log10) was used for modelling the variance.

Generalised linear mixed models (GLMMs) were used to estimate the per-cafeteria effects of the interventions on subsets of the data (with an adjustment for subgroup testing) due to the markedly different patterns at each cafeteria and the quantity of data not allowing an overall additive model to be used. Day of the week was used as a random term, and day number during each intervention was used to allow for 3 separate time trends.

To evaluate the difference in impact between the 2 interventions rather than baseline, the model was rerun with availability instead of baseline used as the reference group leading to the same model fit. Model diagnostics were assessed using variance inflation factors, residual plots, and quantile–quantile (Q–Q) plots for GLMM and additive models, and, also, worm plots and correlation function plots for the additive models, and these diagnostics were acceptable. Further analyses describing formal tests of replication and moderators can be found in S1 Text.

## Results

### Implementation

The fidelity checks suggested that the availability and size interventions were mostly implemented as intended. The overall success rate at achieving the availability targets was 76%. This was calculated by determining the proportion of intervention categories that met the availability intervention targets at each site visit for each cafeteria. Among intervention category products, there were 58% higher energy products available in cafeterias during baseline, which decreased to 50% during availability and 47% during availability plus size. The success rate of implementing the size intervention was 83%. This was calculated by determining the proportion of intervention categories that met the size intervention targets at each site visit for each cafeteria. During the size intervention, 7% of intervention category products being sold during a typical day were reduced in size by an average of 14%. The proportion of products changed was higher for main meals (25% [0.8/3.4 main meals per day]) and sides (42% [2.8/6.7 sides per day]), the categories from which most energy was purchased. See Table 2 and S1 Text for further details.

### Primary outcome

The effect of the availability intervention was a reduction in energy purchased, relative to baseline, of −4.8% (95% CI −7.0% to −2.7%), $p < 0.001$. The effect of the availability plus size intervention, relative to baseline, was a reduction of −11.5% (95% CI −13.7% to −9.3%), $p < 0.001$. These results are shown in Fig 3, and the unadjusted data are shown in Table 3.

The effect of the availability plus size intervention relative to the availability intervention was a reduction in energy purchased of −6.6% (95% CI −7.9% to −5.4%), $p < 0.001$.

The full models (reported in S1 Text) also show nonsignificant overall time trends for baseline, $B = -1,841.35$, $SE = 1,605.49$, $p = 0.252$; availability, $B = -174.31$, $SE = 467.23$, $p = 0.709$; and availability plus size, $B = -157.38$, $SE = 244.14$, $p = 0.519$, suggesting that on average, the effects of each intervention are maintained over time.

The direction of effects at the cafeteria level was a numerical reduction in energy purchased for 28 out of 36 possible hypothesis tests. However, following a Bonferroni adjustment for subgroup testing ($\alpha < 0.000877$), the availability intervention only significantly reduced energy in

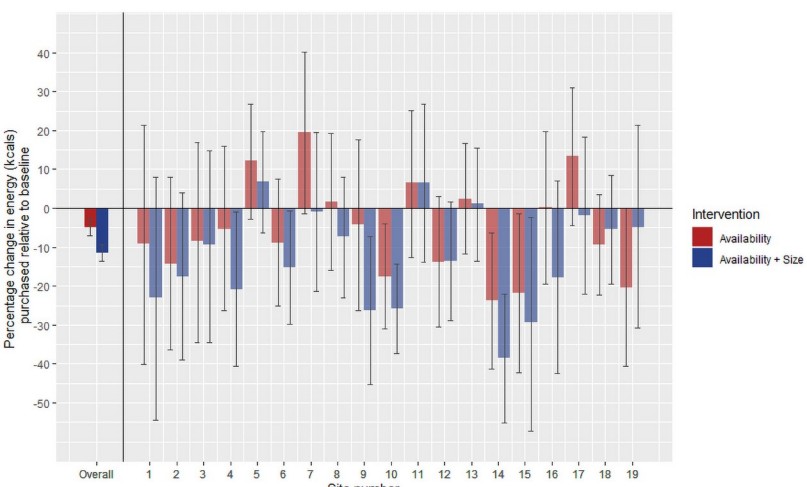

**Fig 3. The effects of availability and availability plus size on energy (kcals) purchased per day relative to baseline.**
Error bars represent 95% CIs for the overall effects and 99.9% CIs (Bonferroni adjustment) for the site effects.

3 cafeterias (cafeteria 10: −17.5% [95% CI −31.0% to −4.1%]; cafeteria 14: −23.8% [95% CI −41.2% to −6.3%]; and cafeteria 15: −21.7% [95% CI −42.3% to −1.4%]). The availability plus size intervention reduced energy purchased in 6 cafeterias using the same adjustment (cafeteria 4: −20.8 [95% CI 40.6% to −1.0%]; cafeteria 6: −15.3% [95% CI −29.8% to −0.8%]; cafeteria 9: −26.2% [95% CI 45.3% to −7.3%]; cafeteria 10: −25.8% [95% CI −37.4% to −14.3%]; cafeteria 14: −38.6% [95% CI −55.1% to −22.1%]; and cafeteria 15: −29.4% [95% CI 57.2% to −2.3%]).

## Secondary outcomes

**Energy purchased from nonintervention categories.** The overall effect of the availability intervention on energy purchased from nonintervention categories relative to baseline was a reduction of −10.1% (95% CI −13.1% to −7.1%), $p < 0.001$. The effect of the availability plus size intervention relative to baseline was a reduction of −9.7% (95% CI −12.9% to −6.5%), $p < 0.001$. There was no meaningful effect of the availability plus size intervention relative to the availability intervention, 0.4% (95% CI −7.7% to 8.6%), $p = 0.808$. However, there were a number of unexplained outliers and erratic time trends in these data, so we recommend caution in interpreting these results.

The time trends from the full models show decreases in energy purchased throughout baseline, $B = -5,836.63$, $SE = 1,775.54$, $p = 0.001$; however, the time trends throughout availability,

**Table 3. Unadjusted data (mean [SD]) for purchases, revenue, and prices in cafeterias during intervention periods.**

|  | Baseline | Availability | Availability plus size |
|---|---|---|---|
| Energy (kcal) purchased per day, per cafeteria from intervention categories | 103,705 | 104,447 | 100,061 |
| Number of transactions per day, per cafeteria | 248 | 275 | 285 |
| Revenue (£) per day, per cafeteria | £598.35 | £582.38 | £610.90 |
| Price per product | £0.91 (£0.42) | £0.92 (£0.39) | £0.88 (£0.39) |

Note: The unadjusted energy purchased is marginally higher in the availability period relative to baseline—in contrast to the modelled effect. This is primarily due to the increased number of transactions that took place during the availability period.

$B = -405.19$, $SE = 1,746.36$, $p = 0.817$, and availability plus size, $B = 680.63$, $SE = 1,946.62$, $p = 0.727$ were both nonsignificant.

**Energy purchased on all categories (intervention and nonintervention).** The overall effect of the availability intervention relative to baseline on total energy purchased was a reduction of −4.8% (95% CI −6.6% to −2.9%), $p < 0.001$. The effect of the availability plus size intervention relative to baseline was a reduction of −8.6% (95% CI −10.3% to −6.9%), $p < 0.001$. The effect of the availability plus size intervention relative to the availability intervention was a reduction in energy purchased of −3.8% (95% CI −5.2% to −2.4%), $p < 0.001$. However, a subset of these data—the energy purchased from nonintervention items—has the same issues described above, and, therefore, we also recommend caution in interpreting these results.

The overall time trends from the full models were nonsignificant throughout baseline, $B = -4,069.70$, $SE = 2,200.78$, $p = 0.065$; availability, $B = -2,593.99$, $SE = 2,163.05$, $p = 0.231$; and availability plus size, $B = 3,046.76$, $SE = 2,418.44$, $p = 0.208$.

**Revenue.** The overall effect of the availability intervention relative to baseline on total revenue was a reduction of −2.6% (95% CI −4.3% to −0.9%), $p < 0.001$. The effect of availability plus size relative to baseline was a reduction of −5.7% (95% CI −7.3% to −4.0%), $p < 0.001$. The effect of availability plus size relative to availability was a reduction in revenue of −3.2% (95% CI −4.9% to −1.4%), $p < 0.001$.

The full models show nonsignificant time trends for baseline, $B = 2.44$, $SE = 6.02$, $p = 0.685$ and decreases in revenue throughout the availability period, $B = -19.48$, $SE = 6.24$, $p = 0.002$. However, there were significant increases throughout the availability plus size period, $B = 35.13$, $SE = 6.93$, $p < 0.001$, despite the overall drop in revenue during this period relative to baseline and availability. This suggests that, although the interventions may reduce revenue, there is uncertainty regarding how long these drops in revenue would last.

## Discussion

The current study evaluated 2 interventions that altered the food environments in 19 worksite cafeterias located throughout England, Scotland, and Wales. Decreasing the relative availability of higher energy foods and reducing their portion size both reduced energy purchased in worksite cafeterias. These interventions have the potential to meaningfully reduce the energy people purchase in cafeterias, particularly when implemented together.

The current study builds upon and replicates the evidence generated from earlier smaller studies [14,15]. The worksites where the cafeterias were located were predominantly staffed by those working in manual occupations. Those in manual occupations have—on average—worse health outcomes and higher body mass indexes (BMIs) compared to those in nonmanual occupations [7], and many interventions that aim to improve healthy eating only exacerbate existing inequalities [20]. The effectiveness of the availability and size intervention in the current sample therefore indicates the potential to improve health equitably, assuming that the interventions are not more effective among those in managerial or higher socioeconomic status groups. This current study also strengthens the conclusions reached in Cochrane reviews regarding the effectiveness of availability and size interventions [12,13]. Existing studies have only tested these interventions in isolation [14,15] or combined them in multicomponent interventions without determining the independent contribution of each [21,22]. The current study builds on this earlier work, demonstrating that adding the size intervention to the already present availability intervention reduces energy purchased even further. The current study took place over a longer time period than these earlier studies, and there was no evidence that the intervention impact was diminished with time. Sustained behaviour change is a major

obstacle to reducing BMI [23,24], and the current results suggest that the 2 evaluated changes to the food environment provided sustained change over the 12 to 21 weeks of intervention.

It should be noted, however, that there was variation in outcomes between the different cafeterias. We observed statistically significant effects in only 3 of the 19 cafeterias for availability and 6 of the 19 cafeterias for availability plus size. This is due partly to the conservative adjustment that was used for multiple testing. It can also be explained by the limited power that a single site has for detecting a change in energy purchased. The extent to which the interventions were implemented, which varied across cafeterias, will also have contributed to different effect sizes. Moderation analyses reported in S1 Text provide mixed support for this latter interpretation. However, the power to test for any moderation effects was very limited. The cafeterias differed in many other ways including the region of the country in which they were based and the demographic characteristics of the employees. These and other unknown factors may also have contributed to variations in effects across sites. It should also be noted that effects were evident in nonintervention products such as breakfasts and sandwiches, an effect which was not observed in earlier studies [14,15]. As the data for the nonintervention category were strongly influenced by outliers and erratic time trends, it is therefore unclear whether this apparent effect on nonintervention products is robust and would need replicating before conclusions can be made. Thorough examination of the data and discussions with site managers did not detect the cause of these data characteristics. However, the same checks were conducted for all other outcomes and were not found to influence the primary outcome or the revenue outcome. Despite this, one possibility is that the interventions changed norms about healthier food choice and portion sizes that changed behaviour even in categories that were unchanged by the interventions [25,26].

The current interventions resulted in a drop in revenue for the cafeterias, unlike earlier studies that did not observe any change in revenue [14]. This may be a temporary effect, as time trends suggested that this drop decreased over the duration of the availability plus size intervention. The observed drop in revenue in the current study may in part be due to the fixed menu and product list that were required for the study. If implemented in practice, cafeterias could respond to lower sales by altering which products are sold and via additional strategies to make healthier food options more appealing [27]. Such additional changes were not permitted in the current study to ensure the accuracy of the energy content across products and to minimise the reprogramming of till buttons mid-study.

## Strengths and limitations

The main strengths of this study are its design and execution. Although there was variation, every site implemented the availability and size interventions to some extent, and a large majority met the prespecified targets. The primary and secondary outcomes in the current study were measured objectively using purchasing data from electronic point-of-sale tills. The company that provided the sales data were unable to provide individual-level data due their data privacy rules. We therefore do not know the extent to which different transactions represented different customers. While our primary outcome was energy purchased—not energy consumed—people tend to consume approximately 90% of food that they select in similar settings [28,29], which suggests that sales are good proxy for consumption. There have also been concerns that such interventions could lead to compensatory consumption such as by purchasing more snacks later in the day; experimental studies find that reducing the portion size of meals does not have this effect [30–33]. It is unknown if compensation effects did occur in our study as we did not test for this; however, the net effect of the interventions was a reduction in energy purchased. There were further concerns by catering staff that the changes would

not be acceptable to customers. However, once the study started, very few comments or complaints were brought to our attention. The availability intervention involved replacing some higher energy products with lower energy ones. While this is conceptually distinguished from an intervention in which the placement of items is systematically altered, the act of removing and replacing products may also change the placement of the products on sale—with sites asked to place healthier products where less healthy ones were previously. While changing the order of products may affect food selection and purchasing [12], it is unclear whether changes to placement necessitated in the availability intervention could have contributed to the effect observed, particularly given variation in site layout and displays usually featuring a mix of the healthier versus less healthy options within the study categories investigated.

We discovered evidence that till buttons were occasionally being used to sell products other than the button's designated product despite instructions and training provided to the catering staff. A sensitivity analysis that adjusts for button presses that were identified as incorrect showed similar patterns of results, yet with altered effect sizes.

## Implications for policy and practice

The current study provides evidence that these changes are effective in a setting primarily used by manual workers, so—providing these are not found to be disproportionately more effective for nonmanual workers—the current interventions would be unlikely to exacerbate and could reduce existing health inequalities [6,34].

A review of energy overconsumption suggests that a reduction of only 28 kcals consumed per person, per day, would be sufficient to prevent further weight gain in 90% of the population [35]. Therefore, if cafeterias in worksites, schools, and universities implemented these changes, this could help reduce overconsumption of energy and therefore aid in widespread efforts to reduce population-level obesity equitably. UK government guidelines for catering in public sector institutions already include best practice guidelines, which include availability and size criteria for snacks and for sugar-sweetened beverages [36]. Extending these guidelines to include recommendations for other products is now warranted, particularly for main meals that contribute the largest share of calories to food purchases in worksite cafeterias.

Those intending to implement these policies should take into consideration the number of products targeted and the extent to which they are reduced in size, as greater reductions in size should lead to greater reduction in energy consumption [37]. While challenging to implement, applying the changes to more products would limit the ability of customers to select the remaining higher energy options. The availability intervention involved changing the proportion of higher energy products from 58% to 50%. Although it is likely that making greater changes than this would lead to greater reductions in energy purchasing and consumption, there is no direct evidence yet to confirm this hypothesis.

## Implications for research

The 3 key uncertainties that remain concern the long-term maintenance of effects, the optimal characteristics of the interventions, and the specific contexts in which the interventions are most effective. First, although the current study was the largest and longest of its kind (to our knowledge), further evaluations are needed to confirm that the interventions will continue to exert their effects over a year or more. Second, the current study did not test the optimal degree of implementation. Future research could estimate the effect of experimentally varying the degree of implementation, particularly for availability interventions, where less research has been conducted. Third, the current study shows that the interventions are effective in worksite cafeterias with predominantly manual workers, which extends further studies showing

effectiveness in cafeterias catering to a mixture of office and manual workers [14,15], in university cafeterias [38], and in a childcare centre [39]. Further research in other out-of-home settings, such as fast food outlets and restaurants, would determine whether there are limits to, or key characteristics associated with, the settings in which the interventions are effective.

## Conclusions

Replacing some higher energy foods in cafeterias with lower energy options and reducing the portion size of some higher energy foods are both effective strategies for reducing energy purchased in worksite cafeterias, particularly in combination. These interventions can contribute to broader strategies to reduce energy intake out of the home, as part of national and international efforts to tackle overweight and obesity.

## Supporting information

**S1 Text. Table A**: Criteria for categorising higher and lower energy foods. **Table B**: Energy purchased per day from intervention categories: full models. **Table C**: Energy purchased per day from nonintervention categories: full models. **Table D**: Energy purchased per day from all categories: full models. **Table E**: Revenue per day from all categories: full models. **Table F**: A comparison of the primary analysis and sensitivity analysis for the primary outcome. **Table G**: Table of correlations between intervention implementation and effectiveness.
(DOCX)

**S1 CONSORT Checklist. Checklist of information to include when reporting a SW-CRT.** CONSORT, Consolidated Standards of Reporting Trials; SW-CRT, stepped-wedge cluster randomised trial.
(PDF)

**S1 Data. Intervention characteristics.**
(XLSX)

**S2 Data. Intervention characteristics.**
(XLSX)

## Acknowledgments

We are grateful to everyone involved in running this project including representatives from the supermarket group, the 3 catering providers, the company that maintains the till systems, and the members of staff at every cafeteria. We also are grateful to Natasha Maynard for assistance and advice during the trial and to Eleni Mantzari, Anna Blackwell, and Rebecca Richards for critical comments on an earlier version of the manuscript.

## Author Contributions

**Conceptualization:** Gareth J. Hollands, Theresa M. Marteau.

**Data curation:** James P. Reynolds, Minna Ventsel, Daina Kosīte, Brier Rigby Dames, Laura Brocklebank, Sarah Masterton, Emily Pechey, Mark Pilling, Rachel Pechey.

**Formal analysis:** Mark Pilling.

**Funding acquisition:** Rachel Pechey, Gareth J. Hollands, Theresa M. Marteau.

**Investigation:** James P. Reynolds, Minna Ventsel, Daina Kosīte, Brier Rigby Dames, Laura Brocklebank, Sarah Masterton, Emily Pechey.

**Methodology:** James P. Reynolds, Mark Pilling, Gareth J. Hollands, Theresa M. Marteau.

**Project administration:** James P. Reynolds, Minna Ventsel, Daina Kosīte, Brier Rigby Dames, Laura Brocklebank, Sarah Masterton, Emily Pechey, Rachel Pechey.

**Supervision:** Rachel Pechey, Gareth J. Hollands, Theresa M. Marteau.

**Visualization:** James P. Reynolds.

**Writing – original draft:** James P. Reynolds.

**Writing – review & editing:** James P. Reynolds, Minna Ventsel, Daina Kosīte, Brier Rigby Dames, Laura Brocklebank, Sarah Masterton, Emily Pechey, Mark Pilling, Rachel Pechey, Gareth J. Hollands, Theresa M. Marteau.

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
