## [Editor Report · Decision Letter 0]

24 Dec 2020

Dear Dr Reynolds, 

Thank you for submitting your manuscript entitled "Impact of decreasing the proportion of higher energy foods and reducing portion sizes on energy purchased in worksite cafeterias: a stepped-wedge randomised controlled trial" for consideration by PLOS Medicine.

Your manuscript has now been evaluated by the PLOS Medicine editorial staff and I am writing to let you know that we would like to send your submission out for external assessment.

Please re-submit your manuscript after the holidays.

Once your full submission is complete, your paper will undergo a series of checks in preparation for external assessment.

Kind regards,

Richard Turner, PhD

rturner@plos.org

---

## [Decision Letter · Decision Letter 1]

10 Mar 2021

Dear Dr. Reynolds,

Thank you very much for submitting your manuscript "Impact of decreasing the proportion of higher energy foods and reducing portion sizes on energy purchased in worksite cafeterias: a stepped-wedge randomised controlled trial" (PMEDICINE-D-20-06163R1) for consideration at PLOS Medicine. 

Your paper was discussed with an academic editor with relevant expertise and sent to independent reviewers, including a statistical reviewer. The reviews are appended at the bottom of this email and any accompanying reviewer attachments can be seen via the link below:

[LINK]

In light of these reviews, we will not be able to accept the manuscript for publication in the journal in its current form, but we would like to invite you to submit a revised version that addresses the reviewers' and editors' comments fully. You will appreciate that we cannot make a decision about publication until we have seen the revised manuscript and your response, and we expect to seek re-review by one or more of the reviewers. 

We hope to receive your revised manuscript by Mar 31 2021 11:59PM. Please email us (plosmedicine@plos.org) if you have any questions or concerns.

Please let me know if you have any questions, and we look forward to receiving your revised manuscript. 

Sincerely,

Richard Turner PhD

Senior editor, PLOS Medicine

rturner@plos.org

To your data statement, please add details for a non-author contact for readers who may be interested in inquiring about access to the study data.

Please avoid "energy purchased" in the title, perhaps with a different form of words.

Please combine the "Methods" and "Findings" subsections of your abstract. Please add a new final sentence to the combined subsection, which should begin "Study limitations include ..." or similar and quote 2-3 of the study's main limitations. 

Please quote study dates in the abstract.

Please mention the unit of randomization in the abstract; and list the relevant secondary outcomes.

After the abstract, we will need to ask you to add a new and accessible "Author summary" section in non-identical prose. You may find it helpful to consult one or two recent research papers published in PLOS Medicine to get a sense of the preferred style. 

Could "written consent", mentioned in the Methods section, be described as "informed"?

Please rename figure 1 "Study site flowchart" or similar.

Please add an additional sentence or two to the first paragraph of your Discussion, so as to summarize the study and its findings. 

Please avoid claims such as the "largest" or "first", as in the Discussion section, and where needed add "to our knowledge" or similar. 

Throughout the text, please adapt reference call-outs to the following style: "... food and drinks [4,5].".

Please remove the information on data sharing, funding and competing interests from the end of the main text. In the event of publication, this information will appear in the article metadata via entries in the submission form. 

Please remove all iterations of "[Internet]" from your reference list.

Noting reference 11, please ensure that all references include full access details. 

Please rename the attached CONSORT checklist "S1_CONSORT_Checklist" or similar, and refer to it by this label in your Methods section. Please adapt the checklist so that individual items are referred to by section (e.g., "Methods") and paragraph number rather than by line or page numbers, as the latter generally change in the event of publication. 

Comments from the reviewers:

*** Reviewer #1: 

The authors test large-scale interventions decreasing availability of 'higher energy' products. There sample consisted especially of male staff working in manual occupation labor force. Their results show a decrease of higher energy foods sold in cafeterias when the proportion of these higher energy foods was decreased and when portion sizes were reduced. The paper is well written and gives a clear overview of how the interventions have been implemented and the results of implementing these interventions. I have only minor comments/questions. 

* During intervention period no recipes could be altered or meals could be changed, while this was the case prior to the study. What happened when an ingredient fell short during the intervention period? Was the ingredient replaced? Was the product taken away from the assortment? 

* Was there a logbook with notes on problems for example, when one ingredient fell short, when the till didn't work?

* How were the dished placed? Were 'more healthy' products placed between the other products or placed at a specific place (for example, offered at the beginning of the counter?). Placement of dishes everywhere the same? Refer to research on where dishes are placed affect purchases. 

* When taking away the "more energy" products in the availability intervention, did you replace the products highest in energy? 

* During the availability intervention the aim was to have about 50% high energy of most type of meals (e.g. cooked main meals) while for snacks this was 75 or 80%. Why? Are there not enough lower energy snacks? 

* Were there control cafeterias in which no interventions were implemented so that you could control for larger fluctuations in meal choice based on for example public policy campaings, news issues, seasonal fluctuations? 

* Was the reduction in size observable by the participants? The authors argue that there probably were no compensatory effects. Do you have individual tickets to look at whether more side dishes were chosen or more desserts when the size and/or availability intervention was implemented? 

* Will health posters as such already affect behavior (intentions? Accessibility health?) were these posters displayed during the baseline period? 

* Participants do not have to put the food on the plate themselves (i.e. choosing how many spoons they take)? So the meals were served to them? 

* Can you control for out of stock of higher energy meals? If one wants to choose a high energy meal but it is out of stock, one probably chooses a low energy meal. Were the high energy meals often out of stock or chosen first? This could mean that people still had the intention to eat high energy meals but were restricted by the fact that the product was out-of-stock (which could decrease the chances of building a "habit" of choosing less high energy products). 

* Are participants aware of the energy content of the dishes, is this labeled on the products? 

* Did participants complain about a change in assortment or sizes? 

* There were more transactions in the availability and availability + size condition versus the baseline. Can you find out whether these were unique customers or customers that bought a meal and then later on went back to buy another meal? 

* Could the fact that people also buy less high energy products in the non-intervention categories indicate that you created a more healthy mindset by for example the poster or by choosing or seeing more 'less energy products' in the intervention categories? 

* One of the supplementary materials was password protected so could not be opened "excel document "availability characteristics by site")

*** Reviewer #2: 

This is a statistical review of manuscript PMEDICINE-D-20-06163R1 reporting the results of a stepped-wedge randomised trial assessing the impact of decreasing the proportion of higher energy foods and reducing portion sizes on energy purchased in worksite cafeterias. Overall, I found the manuscript clear and easy to follow and only have minor comments listed below.

* Abstract: Please add the number of cafeterias (19) and number of groups/wedges (10) in the methods paragraph.

* I find Figure 1 difficult to understand, partially due to potential confusion between steps and groups/wedges. To make it clearer, I would suggest relabelling the boxes "Availability" and "Availability+Size" as Availability (Step 1)" and "Availability+Size (Step 2)" respectively and, in the 20 boxes, simply indicate the week when the corresponding intervention was started e.g. "Week 5" and "Week 13" for Group 1.

* Figure 2 is a great way to visualise the overall design.

* In the sample size calculation section, please indicate the corresponding design (number of steps and number of groups) as well as the intra cluster correlation assumed.

* The primary outcome is the total energy (kcal) sold by a cafeteria per day which does not take into account potential variations in the number of "clients". The list of covariates includes number of transactions per day as a proxy for the number of customers; however, the methods do not appear to describe covariate adjustments. Please add a description of the full models and included covariates in the methods. Instead of adding the daily number of transactions as a covariate in the model, I wonder whether it might be useful to model the average energy per transaction (i.e. the ratio of total daily energy divided by daily number of transactions) as a separate outcome.

* Please clarify the distribution and link function used in the model e.g. normal distribution with identity link.

* I note that the effect of the intervention is reported as a relative difference (%) together with its confidence interval instead of a mean difference in energy. Please clarify how these quantities were derived from the models. Were log transformations used?

* A 2-sided 95% confidence interval is reported for the primary outcome (see top of Page 15); however, the power calculation mentions a one-sided test with 5% which would correspond to a 2-sided 90% confidence interval (5% on each side). Please clarify/correct. 

* When reporting the results, including tables and figures, please indicate that the outcome is the energy purcuhased daily.

* In Table S2, please clarify what the estimates represent including the exact units. Please also clarify units related to covariates. For example, does the estimate of 36.77 for temperature indicate that for each unit increase in temperature (one degree Celsius?), the total daily energy purchase increases by 36.77 kcal?

* The full model includes a linear time trend for each period (baseline, availability, availability plus size). These effects assume a linear trend with day as the unit of time. Has the assumption of linearity been checked? One alternative could be to model time trends based on the study week (1 to 25) instead of day and without assuming a specific relationship (e.g. linear). This could be done by adding week as a class/categorical effect across the entire study period. Has this been considered?

* Please consider reporting at most one decimal when reporting beta coefficients for absolute changes in kcal (cf Table S2 for example).

* Please consider including a plot of the average daily energy purchased over time with the day within each period on the x-axis.

* In place of the individual cafeteria estimates presented in the supplement, please consider using visual representation showing daily or weekly estimates over time by cafeteria.

* Please indicate whether missing data was present and, if so, what methods were applied.

* According to Table 3, the energy purchased per day increased during the availability period compared to the baseline period; however, the results from the models suggest the opposite direction. I understand that the modelled estimates are subject to adjustments (e.g. due to temporal trends or random effects); however, some explanation including potential plots could help understand this apparent discrepancy. 

* Given than Table 3 shows raw/unadjusted estimates, I would suggest presenting it before presenting the results from the models.

* From my perspective, the primary outcome appears to be more a measure of intervention fidelity rather than a measure of impact (e.g. weight loss, change in diet, etc.). I would suggest adding this as a limitation.

-Laurent Billot

*** Reviewer #3: 

Thank you for the opportunity to review this manuscript. The present paper examines the effect of changing the availability and portion sizes of higher energy foods in 19 worksite cafeterias used by 20,327 employees on the energy purchased. The study is very impressive in terms of its methodological rigorousness, sample size and it is addressing a timely topic. My comments follow:

The introduction should flesh out more the conceptual and theoretical framework. The TIPPME framework is only briefly mentioned in the methods section. However, the classification of "availability x product" and "size x product" is very hard to understand without additional information. It seems likely that not all readers are familiar with the TIPPME typology and why and how this might be different from nudging. 

Please briefly elaborate the reasons for and implications of the implemented intervention sequence in the introduction and discussion. The "availability" intervention was introduced first followed by the combination "availability & portion size reduction". Why was a "availability" intervention singled out but not a "portion size" intervention?

How are the 20 steps in Figure 1 related to the 25 week numbers in Figure 2? In order to harmonize Figure 1 and 2, (if possible), I suggest to include the weekly periods instead of steps in Figure 1.

The results show only for 3 out of 19 cafeterias a significant "availability" effect (site 10, 14, 15) and for 6 out of 19 cafeterias a significant "availability & portion size reduction" effect (site 10, 14, 15 and 4, 6, 9). Why were these sites more successful? I suggest to elaborate more on this important aspect in discussion. Are they belonging to the same catering company? Is it related to the size of the assortment or cafeteria?

There are some sites which show an increase in energy purchased (e.g. site 5, 11). Are there any observations why this (counterintuitive) effect occurred?

Concerning the secondary outcomes, "unexplained outliers and erratic time trends" were observed. What are potential explanations for this? Is there a relation to the primary outcomes? I suggest to elaborate this aspect more in the discussion. 

p. 18, line 18; it is stated that there were "large" interventions effects which ranged between a -4.8% to -11.5% reduction in energy purchased, relative to the baseline. What is the metric for classifying these as a "large" effect? Conversely, on p. 19, line 15; it is stated that the interventions resulted in a "small" drop in revenue. However, the drop was between -2,6% to 5,7%. 

The discussion is partly repeating the introduction and the results (e.g., p. 18, line 23ff).

***

[LINK]

---

## [Decision Letter · Decision Letter 2]

16 Jun 2021

Dear Dr. Reynolds,

Thank you very much for submitting your revised manuscript "Impact of decreasing the proportion of higher energy foods and reducing portion sizes on food purchased in worksite cafeterias: a stepped-wedge randomised controlled trial" (PMEDICINE-D-20-06163R2) for consideration at PLOS Medicine. 

Your paper was discussed among the editors and evaluated by our academic editor and one referee. The reviews are appended at the bottom of this email and any accompanying reviewer attachments can be seen via the link below:

[LINK]

We will not be able to accept the manuscript for publication in the journal in its current form, but we would like to invite you to submit a further revised version that addresses the reviewers' and editors' comments fully. You will recognize that we cannot make a decision about publication until we have seen the revised manuscript and your response, and we may seek further external assessment. 

We hope to receive your revised manuscript within two weeks. Please email us (plosmedicine@plos.org) if you have any questions or concerns.

Please let me know if you have any questions, and we look forward to receiving your revised manuscript. 

Sincerely,

Richard Turner, PhD

rturner@plos.org

Noting PLOS' data policy, https://journals.plos.org/plosmedicine/s/data-availability, our firm view is that a solution needs to be found which provides a route for interested readers to access study data that does not involve contacting the article authors. We suggest that making an anonymized dataset available along with the paper or in a publicly accessible repository is the best way to achieve this, but are happy to discuss the matter further as needed. 

Please adapt the final sentence of the "Methods and findings" subsection of the abstract so that this discusses only study limitations. 

Please remove the statement about ethics and consent from the end of the main text, as this should be in the Methods section if not already present there.

Please use the general style "... week 5" throughout the text, although numbers should be spelt out at the start of sentences. 

Please adapt reference call-outs so that there are no spaces within the square brackets (e.g., "...drinks [4,5]."). 

Please remove footnotes (short explanations can perhaps be included in the text). 

Please adapt reference citations so that 6 author names are listed, followed where appropriate by "et al.".

Please add "U S A" to reference 38, and ensure that journal names are abbreviated consistently.

Comments from the reviewers:

*** Reviewer #2: 

I thank the authors for addressing my comments. My only remaining comment relates to the power calculation.

I understand the pragmatic nature of the sample and the fact that the number of cafeterias was more or less fixed; however, one should still report the effect size detectable by the design for a certain level of power. Given this is a cluster (stepped-wedge) design, one should also take the ICC in consideration. If the ICC is unknown, it would be interesting to know the effect size detectable under reasonable assumptions about the ICC. From my perspective, it would also make sense for the power calculation to be redone assuming a 2-sided test of 5% to match the analysis planned and conducted.

-Laurent Billot

***

[LINK]

---

## [Editor Report · Decision Letter 3]

13 Jul 2021

Dear Dr. Reynolds,

Thank you very much for re-submitting your manuscript "Impact of decreasing the proportion of higher energy foods and reducing portion sizes on food purchased in worksite cafeterias: a stepped-wedge randomised controlled trial" (PMEDICINE-D-20-06163R3) for consideration at PLOS Medicine.

I have discussed the paper with editorial colleagues and I am pleased to tell you that, provided the remaining editorial and production issues are dealt with, we expect to be able to accept the paper for publication in the journal.

[LINK]

Please let me know if you have any questions, and we look forward to receiving the revised manuscript shortly.   

Sincerely,

Richard Turner, PhD

rturner@plos.org

Requests from Editors:

To the segment of the title preceding the colon, please add "... in a UK supermarket" or similar. 

In the author summary, please use the active voice in at least one point, e.g., "We carried out a randomised ...".

In the penultimate point, please soften the wording, e.g., "Replacing some higher energy foods ... both appear to be effective strategies for reducing ...".

The current final sentence of the Introduction is really an element of discussion, and we ask you to either move it to the Discussion section or combine it with the preceding sentence (e.g., "So as to estimate the individual and combined effectiveness of availability and size interventions, we designed ..."). 

In the Methods section, was that "written informed consent"?

Numbers can be used at some additional points in the text, e.g., "... 3 of the 19 cafeterias ..." at the end of p.19.

Can the attached CONSORT checklist be adapted to remove "Page no" at the top of the rightmost column?

***

---

## [Editor Report · Decision Letter 4]

27 Jul 2021

Dear Dr Reynolds, 

On behalf of my colleagues and our Academic Editor, Dr Popkin, I am pleased to inform you that we have agreed to publish your manuscript "Impact of decreasing the proportion of higher energy foods and reducing portion sizes on food purchased in worksite cafeterias: a stepped-wedge randomised controlled trial" (PMEDICINE-D-20-06163R4) in PLOS Medicine.

PRESS

Sincerely, 

Richard Turner, PhD 

rturner@plos.org